# Aggression and the Big Five Personality Factors Among Fitness Practitioners and Pre-Workout Consumers

**DOI:** 10.3390/bs14121131

**Published:** 2024-11-25

**Authors:** Alexandru Stefan Cucui-Cozma, Liana Dehelean, Ana-Cristina Bredicean, Ion Papava, Izabela Edina Deverdics, Ana-Maria Cristina Daescu, Cristian Negrea

**Affiliations:** 1Departament of Surgery II, Discipline of Surgery I, “Victor Babes” University of Medicine and Pharmacy, 300041 Timisoara, Romania; cucui.alexandru@umft.ro; 2Departament of Neurosciences, Discipline of Psychiatry, “Victor Babes” University of Medicine and Pharmacy, 300041 Timisoara, Romania; bredicean.ana@umft.ro (A.-C.B.); ana-maria.daescu@umft.ro (A.-M.C.D.); 3Center for Cognitive Research in Neuropsychiatric Pathology (NeuroPsy-Cog), “Victor Babes” University of Medicine and Pharmacy, 300041 Timisoara, Romania; 4“Dr. Victor Popescu” Emergency Military Clinical Hospital, 300080 Timisoara, Romania; 5Department of Psychiatry, Timis County Emergency Clinical Hospital “Pius Brinzeu”, 300736 Timisoara, Romania; izabela.deverdics@hosptm.ro; 6Departament of Kinetotheraphy and Special Motricity, Faculty of Physical Education and Sport, West University Timisoara, 300223 Timisoara, Romania; cristian.negrea@e-uvt.ro

**Keywords:** aggression, personality, Big Five, fitness, pre-workout

## Abstract

**Background**: The use of pre-workout supplements has surged among fitness practitioners, with various ingredients purported to enhance performance and recovery. This study aims to explore the potential link between pre-workout supplement consumption and aggression, as well as the correlation between the Big Five personality traits and aggression levels. **Methods**: The sample comprised 62 male fitness practitioners aged 20–55 years, divided into two groups: 32 pre-workout consumers and 30 non-consumers. Participants were assessed using the Buss and Perry Aggression Questionnaire and the NEO Five-Factor Inventory (NEO-FFI). **Results**: The results indicated no statistically significant differences in aggression levels between the supplement users and the control group. However, notable personality differences were observed, with pre-workout users showing lower Neuroticism and higher Agreeableness and Conscientiousness compared to non-users. Correlation analyses revealed a significant positive relationship between Neuroticism and all forms of aggression, while Agreeableness and Conscientiousness were negatively correlated with aggression. **Conclusions**: These findings suggest that personality traits may play a more substantial role in moderating aggression among pre-workout users than the supplements themselves. Further research is needed to clarify the potential long-term effects of pre-workout supplementation on aggression and personality dynamics.

## 1. Introduction

In recent years, there has been an observed increase in the use of pre-workout supplements among athletes [1]. Pre-workout supplements are a specialized category of dietary supplements intended to be consumed before engaging in physical activities. The use of nutritional supplements to enhance athletic performance has become an increasingly popular strategy among athletes and fitness enthusiasts. The purposes of using pre-workout supplements include improving athletic performance, accelerating recovery, increasing energy, enhancing nutrient intake, promoting weight loss, and increasing muscle mass [2,3].

Dietary supplements are not regulated by the National Agency for Medicines and Medical Devices in Romania, leading to the emergence of numerous commercial formulas on the market. Most pre-workout supplements contain combinations of multiple ingredients, providing a convenient alternative to taking several different supplements before beginning physical activity. Another advantage is the potential enhancement of ingredient effects when used in combination. Despite the significant heterogeneity of these products, multiple studies have been conducted on their efficacy and safety, yielding mixed results [4,5,6]. The lack of regulation allows each brand of pre-workout supplements to select its own ingredients. Most brands share common ingredients such as caffeine, creatine, and various amino acids and vitamins.

Caffeine, a key ingredient found in most pre-workout supplements, is primarily included due to its positive effects on physical endurance. It is rapidly absorbed, reaching peak plasma levels 60 min after ingestion, and acts as an adenosine receptor antagonist. Acute consumption of caffeine has been shown to improve cognition and performance during endurance, strength, and resistance exercises when consumed in doses ranging from 3 to 6 mg/kg body weight. However, higher doses can cause adverse effects such as anxiety, psychomotor agitation, and headaches, which could negatively impact athletic activities and may be associated with increased aggression, according to the International Society of Sports Nutrition (ISSN) [7].

L-tyrosine, a non-essential amino acid synthesized primarily in the liver by hydroxylating phenylalanine, is a precursor to catecholamine neurotransmitters (dopamine, norepinephrine, and epinephrine). Plasma levels of L-tyrosine peak 1–2 h after consumption and can remain elevated for up to 8 h. Elevated levels of tyrosine have the potential to increase brain dopamine and norepinephrine synthesis. Norepinephrine depletion could compromise physical and cognitive performance [8].

Beta-alanine is a non-essential amino acid that increases intramuscular carnosine content (β-alanine-L-histidine), which plays a role in maintaining intracellular pH during high-intensity physical activity, protecting muscle cells against lactic acidosis and thereby ensuring resistance to muscle fatigue. Arginine is a conditionally essential amino acid used as a pre-workout supplement because it is converted into nitric oxide. Nitric oxide has been suggested to enhance physical performance through its vasodilatory properties, thereby increasing blood flow to skeletal muscles, which, in turn, enhances nutrient delivery and the removal of metabolic waste products resulting from physical exertion [6].

Creatine is an amino acid synthesized endogenously and found in mammalian muscles. Supplements containing creatine increase the availability of adenosine triphosphate (ATP) by enhancing phosphocreatine stores in muscles, thereby improving muscle strength during short-duration, high-intensity exercises. Numerous articles have been published detailing the safety and efficacy of creatine supplementation in humans [9,10].

The B-vitamin complex is important for optimal energy supply, muscle tissue repair, erythrocyte production, protein synthesis, and tissue repair [11]. Vitamin C is a water-soluble vitamin that acts as a cofactor and redox catalyst in many biochemical processes and reactions in the human body, such as tissue growth and repair through collagen synthesis, vasodilation through increased nitric oxide synthesis, and the moderation of oxidative stress [12].

In general, pre-workout supplements have been linked to increased aggression, particularly when they contain anabolic steroids [13,14]. In the absence of these compounds, there is no published literature addressing the association between anabolic steroid-free pre-workout supplements and aggression. Most pre-workout supplements contain caffeine, a potent stimulant that can lead to a temporary increase in energy, alertness, and arousal. Although caffeine consumption has often been associated with increased aggression in various forms, particularly among young people, there are no studies in the literature analyzing the potential association between the consumption of caffeine-containing pre-workouts and aggression [15,16].

Personality and aggression have been extensively studied in psychology and related fields. The relationship between personality traits and aggressive behavior is complex, with several factors potentially influencing the connection between personality and aggression [17,18,19]. Certain personality traits have been associated with aggression. For example, traits such as high levels of hostility, impulsivity, low agreeableness, low self-control, and high neuroticism have been linked to a greater likelihood of aggressive behavior. These traits can influence how individuals respond to challenges or stressful situations [19,20].

The aim of the present study was to determine whether the consumption of a pre-workout supplement (free of anabolic steroids), whose composition corresponds to the majority of those available on the market, led to a significant increase in aggression levels and whether certain personality dimensions from the Big Five model are significantly correlated with various forms of aggression, as assessed by the BPAQ scale [21].

## 2. Materials and Methods

### 2.1. Study Design and Participants Demographics

A descriptive cross-sectional observational study was conducted prospectively, aimed at exploring the relationships between pre-workout supplement consumption, aggression levels, and personality traits among fitness practitioners. The study included 62 male fitness practitioners aged between 20 and 55 years. Participants were selected with the assistance of fitness instructors from various gyms in Timișoara. The study was carried out over a period of three months. The participants were divided into two groups: one group of 32 individuals who regularly practiced fitness and consumed pre-workout supplements, and a control group of 30 individuals who practiced fitness but did not consume pre-workout supplements. Participants’ characteristics, including education level, marital status, urban versus rural residence, and employment status, were also collected and analyzed.

### 2.2. Inclusion Criteria and Supplement Selection

Inclusion and exclusion criteria were determined through a brief, structured interview conducted by psychiatrists from the research team. The interview included targeted questions designed to assess participants’ eligibility. Inclusion criteria required that participants had been consuming the pre-workout supplement for at least two months and had adhered to the recommended dosage (one scoop prior to training) up to the time of questionnaire administration. Exclusion criteria included cognitive impairment, a history of substance abuse or dependence on drugs or alcohol, other psychiatric comorbidities, the use of pre-workout products other than the one selected for this study, or an inability to understand the study procedures. The study included only men, as they represented the majority of pre-workout supplement consumers. Women were excluded to avoid the potential confounding effects of gender-related physiological and hormonal differences. This ensured a more homogeneous sample and reduced variability in the results.

The selection of the supplement for this study was based on its widespread use among consumers. One scoop of the supplement contained the following composition: caffeine—200 mg, vitamin C—50 mg, vitamin B3—37.9 mg, vitamin B6—3.6 mg, beta-alanine—750 mg, L-tyrosine—1500 mg, arginine—3000 mg, and creatine—5000 mg.

### 2.3. Psychometric Assessments

All participants completed the Buss and Perry Aggression Questionnaire and the NEO Five-Factor Inventory (NEO-FFI) after their training sessions. The Romanian version of NEO-FFI [22] is a shortened version of the Revised NEO Personality Inventory (NEO-PI-R) developed by McCrae and Costa in 1989 [23]. Following the five-factor model, the basic personality traits assessed by this scale are Neuroticism (the tendency to experience negative emotions such as anxiety, anger, disgust, and depression, along with difficulties in impulse and stress control), Extraversion (the tendency to engage actively in social situations, being communicative, sociable, active, courageous, and self-confident), Openness to Experience (the tendency to be creative, curious, open to adventure, and to experience strong emotions and unconventional ideas), Agreeableness (the tendency to be gentle, altruistic, flexible, tolerant, kind, generous, and honest), and Conscientiousness (the tendency to be organized, disciplined, trustworthy, methodical, but also stubborn and obsessive in certain situations). The questionnaire is self-reported and consists of 60 items, with 12 items corresponding to each personality dimension. Participants rated each statement on a five-point Likert scale, ranging from 1 (“strongly disagree”) to 5 (“strongly agree”). Personality dimension scores were calculated by summing the scores of the items corresponding to the associated subscale. The highest score within a subscale indicates the dominant personality trait [24].

The Buss and Perry Aggression Questionnaire (BPAQ) is one of the most widely used instruments for assessing aggression and consists of 29 items with responses on a Likert scale from 1 (“extremely uncharacteristic of me”) to 5 (“extremely characteristic of me”). This questionnaire measures four dimensions of direct aggression: physical aggression (nine items), verbal aggression (five items), anger (seven items), and hostility (eight items) [21].

All participants provided written informed consent before participating in the study. The study protocol, procedures, and informed consent template were approved by the Ethics Committee of West University of Timișoara.

### 2.4. Statistical Analysis

To summarize and analyze the characteristics of the study population, a descriptive and inferential statistical analysis was performed. The variables included in the study were presented using summary statistics: frequency, mean, standard deviation, median, minimum, and maximum.

To check the normality of the variables’ distribution, the Shapiro–Wilk test was used, with a *p*-value > 0.05 being representative of the Gaussian distribution. Normally distributed (Gaussian) data were analyzed using parametric tests, and non-normally distributed variables were analyzed using non-parametric tests. To assess the difference between groups, the independent samples *t*-test was used for parametric data and the Mann–Whitney U test was used for non-parametric data. Relationships between variables were assessed using Pearson’s correlation coefficient (r) for the Gaussian population and Spearman’s rank correlation coefficient (ρ) for the non-Gaussian population.

Data were collected, processed, and analyzed using “R Core Team (2024). R: A language and environment for statistical computing. R Foundation for Statistical Computing, Vienna, Austria”, and the results were presented in tabular and graphical form. A *p*-value < 0.05 was considered to indicate a statistically significant difference with a 95% confidence interval.

## 3. Results

In the context of examining the relationship between pre-workout supplement consumption, aggression levels, and personality traits among fitness practitioners, the normality of the data was assessed using the Shapiro–Wilk test. The analysis revealed that most variables, including BMI, education level, the Big Five personality traits (Neuroticism, Extraversion, Openness to Experience, Agreeableness, Conscientiousness), and the Buss and Perry Aggression Questionnaire subscales (Physical Aggression, Verbal Aggression, Anger, Hostility), exhibited a normal distribution (*p* > 0.05). Consequently, these variables are suitable for parametric statistical analyses, which allow for the robust exploration of potential relationships and differences.

However, the variables of Age (W = 0.851, *p* = 0.0004) and Duration of Supplement Use (DSU) (W = 0.809, *p* = 0.0001) did not conform to a normal distribution, as indicated by their significant *p*-values (*p* < 0.05). This non-normality suggests the need for non-parametric statistical methods when analyzing these variables, as traditional parametric tests could yield misleading results. The results are presented in Table 1.

In this study, two groups of male fitness practitioners were compared: those who regularly consumed a specific pre-workout supplement and those who did not (control group). The analysis included variables such as age, education level, BMI, personality traits (measured by the NEO-FFI), and various forms of aggression (measured by the BPAQ).

The results showed no significant differences between the groups in terms of age (*p* = 0.88), BMI (*p* = 0.40), or education level (*p* = 0.73), indicating that these demographic variables were comparable. However, notable differences emerged in certain personality traits. The supplement users exhibited significantly lower levels of Neuroticism (*p* = 0.031) and higher levels of Agreeableness (*p* = 0.004) and Conscientiousness (*p* < 0.001) compared to the control group. Although the supplement group also had higher mean scores for Extraversion and Openness to Experience, these differences did not reach statistical significance (*p* = 0.187 and *p* = 0.071, respectively).

When it came to aggression, no significant differences were found between the two groups across all subscales: Physical Aggression (*p* = 0.638), Verbal Aggression (*p* = 0.986), Anger (*p* = 0.507) and Hostility (*p* = 0.091). While the supplement users had slightly higher scores for Hostility, these differences were not statistically significant. In conclusion, while pre-workout supplement users differed significantly from non-users in certain personality traits, these differences did not translate into significant variations in aggressive behaviors, suggesting that personality traits such as lower Neuroticism and higher Agreeableness and Conscientiousness may be more characteristic of supplement users without necessarily leading to increased aggression. The results are presented in Table 2.

The demographic characteristics of the control group and the supplement use group were compared across several categorical variables, including marital status, urban versus rural residence, and employment status. The analysis revealed no significant differences between the groups for any of these variables, as detailed in the following paragraphs and summarized in Table 3.

Regarding marital status, both groups had similar distributions, with a majority of participants being either in a relationship or married. Specifically, 40.625% of the control group and 43.75% of the supplement use group reported being in a relationship, while 37.5% of the control group and 40.625% of the supplement use group were married. The proportion of single and divorced participants was also comparable, with no significant difference observed (*p* = 0.936).

Similarly, the distribution of participants based on urban versus rural residence was almost identical between the two groups, with about half of the participants in each group residing in rural areas (50.0% in the control group and 53.125% in the supplement use group) and the other half in urban areas (50.0% in the control group and 46.875% in the supplement use group). The *p*-value of 0.999 indicates no significant difference in residential status.

Employment status also showed no significant difference between the groups (*p* = 0.915). Most participants in both groups were employed (53.125% in the control group and 56.250% in the supplement use group). The proportions of unemployed individuals, students, and retirees were also similar across the groups.

Overall, these findings suggest that the control and supplement use groups are demographically similar, with no significant differences in marital status, residential status, or employment status. This comparability between the groups supports the validity of further analyses, as it suggests that any differences in outcomes related to supplement use are less likely to be confounded by these demographic factors. The results are presented in Table 3.

The analysis explored the correlations between age, the Big Five personality traits (as measured by the NEO-FFI), and various forms of aggression (as measured by the BPAQ) to examine the interrelationships between these variables. The results revealed several significant associations, highlighting the complex interplay between personality and aggressive behaviors.

Neuroticism was positively correlated with various forms of aggression, including Physical Aggression (r = 0.41, *p* = 0.021), Verbal Aggression (r = 0.36, *p* = 0.045), Anger (r = 0.40, *p* = 0.025) and Hostility (r = 0.42, *p* = 0.015), indicating that higher levels of Neuroticism are associated with increased aggression.

Openness to Experience (O) was negatively correlated with Physical Aggression (r = −0.49, *p* = 0.005), suggesting that those higher in Openness tend to be less aggressive.

Agreeableness (A) also demonstrated significant negative correlations with various forms of aggression. It was strongly negatively correlated with Physical Aggression (r = −0.59, *p* < 0.001), as well as with Verbal Aggression (r = −0.40, *p* = 0.022), Anger (r = −0.40, *p* = 0.025), and Hostility (r = −0.36, *p* = 0.041). These findings indicate that higher Agreeableness is consistently associated with lower levels of aggression.

Conscientiousness (C) exhibited negative correlations with aggression as well, including Physical Aggression (r = −0.42, *p* = 0.017), Verbal Aggression (r = −0.37, *p* = 0.036), Anger (r = −0.38, *p* = 0.032) and Hostility (r = −0.47, *p* = 0.006), suggesting that higher Conscientiousness is linked to lower aggression across the board. The results are presented in Table 4.

## 4. Discussion

In sports, the majority of studies involving aggression have typically focused on parameters such as the location of the sporting event [25], the opposing team [26], the level of competition [27], the frequency of competition [28], and the aggression of the opposition [29]. When considering aggression associated with the consumption of pre-workout supplements, previous evidence has primarily examined this association in the context of supplements containing anabolic steroids [14]. Currently, most pre-workout supplements no longer contain these anabolic substances, instead comprising caffeine and other ingredients such as vitamins and amino acids [5].

In a 2019 study by Ellerbroeck and Antonio, which analyzed, among other aspects, the acute effects of pre-workout supplements on mood dimensions including Tension and Anger, no significant differences were found between the pre-workout supplement users and the placebo group concerning these parameters. However, it is important to highlight that in this study, both the group receiving the pre-workout supplement and the placebo group were administered the same amount of caffeine, as the placebo supplement contained caffeine [30]. Standardizing caffeine intake ensured that any observed differences in dimensions such as Tension and Anger could not be attributed to caffeine alone but rather to the effects of other components in the pre-workout supplement.

Considering the components of the pre-workout supplement, caffeine was the only substance that raised concerns regarding its association with aggression. Numerous studies have mentioned the association between caffeine consumption and aggression, behavioral disorders, and violence [16,31,32]. Caffeine’s influence on aggression occurs through both non-linear effects on aggression [31,33] and through perturbation in the functionality of 5-HT [34], as well as through direct effects via antagonistic action on adenosine 2a receptors [35,36]. In most cases, a positive association has been presented between caffeine consumption and various forms of aggression [37,38].

In various sports (e.g., soccer, rugby, and athletic disciplines), to achieve positive effects, the literature recommends a low to moderate intake of caffeine (3–6 mg/kg) before training sessions [39,40]. A consumption level of 7–10 mg/kg is most frequently associated with possible adverse effects such as palpitations, tremors, headaches, or flushing [41,42], while consumption exceeding 10 mg/kg is accompanied by nervousness, abdominal cramps, insomnia, digestive disorders, and periods of unreasonable alertness, which could further lead to aggression [43,44].

In this study, participants ingested 200 mg of caffeine before training as part of the pre-workout supplement, which falls within the low to moderate caffeine consumption range. Comparing the levels of aggression between the two groups, although higher aggression was observed in the supplement group, the differences were not statistically significant. Therefore, it can be concluded that the appropriate consumption of the supplement used did not lead to a significant increase in aggression compared to those who did not consume the supplement.

Further analyzing the role of personality structure, this study also examined possible correlations between the five personality dimensions of the Big Five model and aggression in the group of pre-workout supplement users. The five-factor model of personality is considered the most prominent model for research [45]. As previously mentioned, the five factors include Agreeableness (altruism, nurturing, conformity, the readiness to maintain positive interpersonal relationships), Conscientiousness (reliability, will to achieve, responsibility, accuracy, and commitment fulfillment), Neuroticism/Emotional Instability (instability of affect, negative thoughts, feelings of anxiety, depression, and irritability), Extraversion (levels of energy, social adaptivity, assertiveness, activity, self-confidence), and Openness to Experience (openness to new ideas, creativity, imagination, intellectual curiosity).

To date, multiple studies have analyzed the relationship between aggression and personality, including the Big Five model [18,46,47,48]. It has been suggested that of the personality dimensions in the Big Five model, Agreeableness and Neuroticism have the strongest associations with aggression/violence. In most studies, Neuroticism has positively correlated with aggression, while Agreeableness has negatively correlated with it [46,49,50]. Thus, Neuroticism has been associated with aggressive behavior [46,50] and aggressive emotions [20], while Agreeableness has been strongly inversely associated with aggression/violence [20,46,48,51,52,53,54].

Regarding the other three personality dimensions, the literature results have been less consistent. For instance, in relation to Extraversion and aggression, previous findings are mixed. In most cases, however, a negative correlation between Extraversion and aggression has been reported [20,50,55]. In the study by Cavalcanti and Pimentel [48], a positive correlation between Extraversion and aggression was reported. Regarding Conscientiousness, previous studies have found either a negative association with aggression [46,48,50,56] or no relationship with aggression [20].

A similar situation is observed in studies examining the relationship between Openness to Experience and aggression. Certain studies have found no relationship between Openness and aggression [20,50,53], while others have reported a positive relationship between Openness and various forms of aggression/violence [48,57].

In the case of the personality dimension Neuroticism, a direct positive correlation was identified with all forms of aggression assessed using the BPAQ scale. This result corresponds with the existing literature, as mentioned earlier, which positively correlates this dimension with aggression/violence [20,46,50]. In other words, the more pronounced this personality dimension is in consumers, the more accentuated the various forms of aggression are in them.

Significant negative correlations were found between the personality dimensions of Agreeableness, Conscientiousness, and all forms of aggression evaluated using the BPAQ scale (physical aggression, verbal aggression, anger, and hostility). Thus, the results of this study align with the majority of studies that have reported a significant negative/inverse relationship between aggression/violence and these personality dimensions [46,48,50,53,54,56]. Therefore, the more pronounced these personality dimensions are, in the case of appropriate pre-workout consumption, the less likely aggression in the analyzed forms is to manifest.

Extraversion did not significantly correlate with any of the forms of aggression evaluated using the BPAQ scale. This result does not align with the findings reported in the literature, which have mentioned either negative correlations (in the majority of cases) [20,51,56] or positive correlations [48]. The inconsistency of these results, along with the result of this study that offers a third variation of this relationship, underscores the need for further studies to reanalyze this relationship.

Regarding Openness to Experience, it was significantly negatively correlated only with physical aggression. This result aligns with the majority of the existing literature that mentions no correlation between this personality dimension and aggression/violence concerning the correlations with verbal aggression, hostility, and anger evaluated using the BPAQ scale [20,50,53]. However, the negative correlation found in this study with physical aggression contradicts the results of studies by Cavalcanti and Pimentel and Jones et al., which described a positive correlation between this dimension and some forms of aggression/violence [48,57]. This result fits within the inconsistency of previous findings regarding the Openness to Experience–aggression relationship.

One limitation of this study is the lack of assessment regarding the use of other supplements (e.g., those for cardio or weight loss) within the control group. Future research should consider controlling for the consumption of non-pre-workout supplements to better isolate the effects of pre-workout formulations.

## 5. Conclusions

The regular and appropriate consumption of a pre-workout supplement, including caffeine within the recommended range of 3–6 mg/kg, did not lead to a significant increase in aggression as measured by the BPAQ scale. The observed correlations between aggression and the Big Five personality dimensions were consistent with the existing literature, except for Openness to Experience, which showed a significant inverse correlation with physical aggression, and Extraversion, which did not correlate significantly with aggression.

These findings suggest that pre-workout supplements, when used as recommended, are safe in terms of aggression, regardless of personality traits. This provides valuable insights for exercisers and trainers, highlighting the importance of adhering to dosage guidelines to minimize risks and optimize performance outcomes.

## Figures and Tables

**Table 1 behavsci-14-01131-t001:** Shapiro–Wilk test results for normality of demographic, personality, and aggression variables.

Variable	Shapiro–Wilk W	*p*-Value	Normal Distribution
Age (years)	0.851	0.0004	No
DSU (months)	0.809	0.0001	No
Education Level	0.969	0.467	Yes
$\mathrm{BMI}\;(\mathrm{kg}/m^{2})$	0.950	0.141	Yes
N (NEO-FFI)	0.967	0.429	Yes
E (NEO-FFI)	0.973	0.575	Yes
O (NEO-FFI)	0.954	0.184	Yes
A (NEO-FFI)	0.967	0.422	Yes
C (NEO-FFI)	0.966	0.406	Yes
PA	0.945	0.101	Yes
VA	0.983	0.877	Yes
A	0.980	0.797	Yes
H	0.965	0.368	Yes

Abbreviations: N—Neuroticism, E—Extraversion, O—Openness to Experience, A—Agreeableness, C—Consciousness, PA—Physical Aggressivity, VA—Verbal Aggressivity, A—Anger, H—Hostility, BMI—Body Mass Index, DSU—Duration of Supplement Use, *p*-value—Shapiro–Wilk test result.

**Table 2 behavsci-14-01131-t002:** Comparison of demographic, personality, and aggression variables between control and supplement use groups.

Variable	Control (N = 30)	Supplement Use (N = 32)	*p*-Value
* Age (years)	27 (7.5)	28.5 (8.5)	0.88
DSU (months)	-	6.0 (9.0)	-
Education Level	10 ± 4	11 ± 3	0.73
$\mathrm{BMI}\;(\mathrm{kg}/m^{2})$	27.08 ± 4.89	28.59 ± 4.66	0.40
N (NEO-FFI)	19.43 ± 6.66	15.40 ± 7.63	0.031
E (NEO-FFI)	28.13 ± 6.29	30.66 ± 5.36	0.187
O (NEO-FFI)	25.23 ± 4.03	27.59 ± 4.27	0.071
A (NEO-FFI)	25.80 ± 4.52	29.38 ± 4.89	0.004
C (NEO-FFI)	29.90 ± 2.45	35.47 ± 6.93	<0.001
PA	19.90 ± 5.31	20.63 ± 6.63	0.638
VA	13.23 ± 3.75	13.25 ± 3.56	0.986
A	16.40 ± 5.50	17.31 ± 5.27	0.507
H	18.23 ± 4.98	20.66 ± 6.02	0.091

* Non-Gaussian distribution variable. Abbreviations: N—Neuroticism, E—Extraversion, O—Openness to Experience, A—Agreeableness, C—Consciousness, PA—Physical Aggressivity, VA—Verbal Aggressivity, A—Anger, H—Hostility, BMI—Body Mass Index, DSU—Duration of Supplement Use, *p*-value—*t*-test/Mann–Whitney U test.

**Table 3 behavsci-14-01131-t003:** Comparison of marital status, urban/rural residence, and employment status between control and supplement use groups.

Variable	Category	Control (N = 30)	Supplement Use (N = 32)	*p*-Value
Marital Status	Single	4 (12.5%)	3 (9.375%)	0.936
	Relationship	13 (40.625%)	14 (43.75%)	
	Married	12 (37.5%)	13 (40.625%)	
	Divorced	3 (9.375%)	2 (6.250%)	
Urban/Rural	Rural	16 (50.0%)	17 (53.125%)	0.999
	Urban	16 (50.0%)	15 (46.875%)	
Employment Status	Employed	17 (53.125%)	18 (56.250%)	0.915
	Unemployed	8 (25.0%)	7 (21.875%)	
	Student	5 (15.625%)	6 (18.750%)	
	Retired	2 (6.25%)	1 (3.125%)	

Abbreviations: *p*-value—Pearson’s Chi-square test.

**Table 4 behavsci-14-01131-t004:** Correlations between age, personality traits, and aggression variables.

Correlation	*p*-Value	*r*	Strength	Association Direction
N (NEO-FFI) and PA	0.021	0.41	Medium	Positive
N (NEO-FFI) and VA	0.045	0.36	Medium	Positive
N (NEO-FFI) and A	0.025	0.40	Medium	Positive
N (NEO-FFI) and H	0.015	0.42	Medium	Positive
O (NEO-FFI) and PA	0.005	−0.49	Medium	Negative
A (NEO-FFI) and PA	0.000	−0.59	Large	Negative
A (NEO-FFI) and VA	0.022	−0.40	Medium	Negative
A (NEO-FFI) and A	0.025	−0.40	Medium	Negative
A (NEO-FFI) and H	0.041	−0.36	Medium	Negative
C (NEO-FFI) and PA	0.017	−0.42	Medium	Negative
C (NEO-FFI) and VA	0.036	−0.37	Medium	Negative
C (NEO-FFI) and A	0.032	−0.38	Medium	Negative
C (NEO-FFI) and H	0.006	−0.47	Medium	Negative

Abbreviations: N—Neuroticism, E—Extraversion, O—Openness to Experience, A—Agreeableness, C—Consciousness, PA—Physical Aggressivity, VA—Verbal Aggressivity, A—Anger, H—Hostility, *r*—Spearman’s Rank Correlation Coefficient, *p*-value—Spearman’s Rank Correlation test result.

## Data Availability

The data presented in this study are available on request from the corresponding author. The data are not publicly available due to reasons concerning privacy of the subjects.

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
