# Peer review of "Aggression and the Big Five Personality Factors Among Fitness Practitioners and Pre-Workout Consumers"

_behavsci, 2024, doi:10.3390/bs14121131_

Round 1

Reviewer 1 Report

Comments and Suggestions for Authors

Research is interesting because it allows for a deeper understanding of the topic under consideration and the discovery of new, unexpected insights. However, I would like to make some comments to improve the quality of the article.

1) It is not indicated in the Materials and Methods section, and therefore it remains unclear whether an experiment was performed or whether it is only a descriptive study (because mentioned control group).

2) When describing the characteristics of the subjects in the Materials and Methods section, the education and other characteristics of the subjects are not mentioned.

3) It is also unclear whether the subjects in the control group used any other (non-pre-work out) supplements, such as for cardio, weight loss, etc. witch could determine the results of the study.

4) A comment is needed as to why only men were chosen for the study.

5) It is interesting why links between the use of supplements and personal characteristics were reached, maybe it would be more useful to link for example with anxiety?

Reviewer 2 Report

Comments and Suggestions for Authors

Thank you for your work on this paper. Below are some suggestions for change:

Line 52: please change "result" to "results"

Lines 54-56: please move this up to the previous paragraph

Lines 57-59: please consider deleting these lines and replacing them with a sub-heading and a transition sentence; please remove "we" language

Lines 69-74: please include a citation

Lines 88-93: please include a citation

Lines 103-106: please include a citation

Lines 108-110: please include a citation

Line 115: please include a citation for the ASQ scale

Please include information about how you determined if someone met inclusion/exclusion criteria.

Please consider streamlining how you share the rationale for supplement composition of the pre-workout supplement. Currently, it's in two sections. After reading the first mention of it, I was unclear. However, after reading the second mention, it became much more clear.

Please include a citation for the definition of the things measured in the 5-factor model.

Line 175; 186; 205: please remove the "we" language

Lines 233-236: please include the appropriate numerical values that are discussed

Line 262: please remove the "our" language

On several occasions throughout the paper, there are one-sentence statements not connected to larger paragraphs. Please review all those and consider whether to add the sentence to the paragraph above or below the statement. 

Line 296: please include a citation for the sentence about "previous evidence"

Line 298: please include a citation

Line 299: please remove "J."

Line 305: please add why it is important to consider that both groups were given the same amount of caffeine

Line 316: Please include what kinds of sports the research says benefit from caffeine consumption. 

Line 322: please remove "our" language

Line 327: please remove "we" language

Line 364-365: Please consider deleting the "some studies have..." sentence. It feels repetitive. 

Line 365: please remove "For example,"

Line 372: please remove "we" language

Line 374: please delete "data"

Line 380; 387; 390: please remove "our" language

Throughout the paper, please check to make sure that in-text citations are uniform in presentation.

Under the "Conclusions" section, please share what the findings mean. What is the take-away the author's wish for readers to have? For example, how might this information impact exercisers and trainers?
